# Multiscale Neural Operator: Learning Fast and Grid-independent PDE Solvers

## Abstract

Numerical simulations in climate, chemistry, or astrophysics are computationally too expensive for uncertainty quantification or parameter-exploration at high-resolution. Reduced-order or surrogate models are multiple orders of magnitude faster, but traditional surrogates are inflexible or inaccurate and pure machine learning (ML)-based surrogates too data-hungry. We propose a hybrid, flexible surrogate model that exploits known physics for simulating large-scale dynamics and limits learning to the hard-to-model term, which is called parametrization or closure and captures the effect of fine- onto large-scale dynamics. Leveraging neural operators, we are the first to learn grid-independent, non-local, and flexible parametrizations. Our *multiscale neural operator* is motivated by a rich literature in multiscale modeling, has quasilinear runtime complexity, is more accurate or flexible than state-of-the-art parametrizations and demonstrated on the chaotic equation multiscale Lorenz96.

## 1 Introduction

Climate change increases the likelihood of storms, floods, wildfires, heat waves, biodiversity loss and air pollution (IPCC, 2018). Decision-makers rely on climate models to understand and plan for changes in climate, but current climate models are computationally too expensive: as a result, they are hard to access, cannot predict local changes ($< 10km$), fail to resolve local extremes (e.g., rainfall), and do not reliably quantify uncertainties (Palmer et al., 2019). For example, running a global climate model at $1km$ resolution can take ten days on a $4888\times$GPU node supercomputer, consuming the same electricity as a coal power plants generates in one hour (Fuhrer et al., 2018). Similarly, in molecular dynamics (Batzner et al., 2022), chemistry (Behler, 2011), biology (Yazdani et al., 2020), energy (Zhang et al., 2019), astrophysics or fluids (Duraisamy et al., 2019), scientific progress is hindered by the computational cost of solving partial differential equations (PDEs) at high-resolution (Karniadakis et al., 2021). We are proposing the first PDE surrogate that quickly computes approximate solutions via correcting known large-scale simulations with learned, grid-independent, non-local parametrizations.

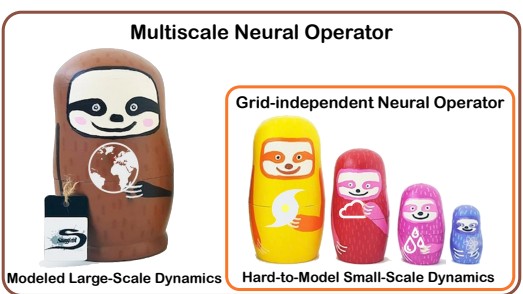

Figure 1: **Multiscale neural operator (MNO).** Explicitly modeling all scales of Earth's weather is too expensive for traditional and learning-based solvers (Palmer et al., 2019). MNO dramatically reduces the computational cost by modeling the large-scale explicitly and learning the effect of fine- onto large-scale dynamics; such as turbulence slowing down a river stream. We embed a grid-independent neural operator in the large-scale physical simulations as a "parametrization", conceptually similar to stacking dolls (Snagglebit, 2022).

Surrogate models are fast, reduced-order, and lightweight copies of numerical simulations (Quarteroni & Rozza, 2014) and of significant interest in physics-informed machine learning (Kashinath et al., 2021; Reichstein et al., 2019; Karpatne et al., 2019; Ganguly et al., 2014). Machine learning (ML)-based surrogates have simulated PDEs up to $1 - 3$ order of magnitude faster than traditional numerical solvers and are more flexible and accurate than traditional surrogate models (Karniadakis et al., 2021). However, pure ML-based surrogates are too data-hungry (Rasp et al., 2020); so, hybrid ML-physics models are created, for example, via incorporating known symmetries (Bronstein et al., 2021; Batzner et al., 2022) or equations (Willard et al., 2022). Most hybrid models represent the solution at the highest possible resolution which becomes computationally infeasible in multiscale or very high-resolution physics; even at optimal runtime (Pavliotis & Stuart, 2008; Peng et al., 2021).

As depicted in Figs. 1 and 2, we simulate multiscale physics by running easy-to-acces large-scale models and focusing learning on the challenging task: *How can we model the influence of fine-onto large-scale dynamics, i.e., what is the subgrid parametrization term?* The lack of accuracy in current subgrid parametrizations, also called closure or residual terms, is one of the major sources of uncertainty in multiscale systems, such as turbulence or climate (Palmer et al., 2019; Gentine et al., 2018). Learning subgrid parametrizations can be combined with incorporating equations as soft (Raissi et al., 2019) or hard (Beucler et al., 2021a) constraints. Various works learn subgrid parametrizations, but are either inaccurate, hard to share or inflexible because they are local (Gentine et al., 2018), grid-dependent (Lapeyre et al., 2019), or domain-specific (Behler J, 2007), respectively as detailed in Section 2. We are the first to formulate the parametrization problem as learning neural operators (Anandkumar et al., 2020) to represent non-local, flexible, and grid-independent parametrizations.

We propose, *multiscale neural operator* (MNO), a novel learning-based PDE surrogate for multiscale physics with the key contributions:

- A learning-based multiscale PDE surrogate that has quasilinear runtime complexity, leverages known large-scale physics, is grid-independent, flexible, and does not require autodifferentiable solvers.
- The first surrogate to approximate grid-independent, non-local parametrizations via neural operators.
- Demonstration of the surrogate on the chaotic, coupled, multiscale PDE: multiscale Lorenz96.

## 2 RELATED WORKS

We embed our work in the broader field of physics-informed machine learning and surrogate modeling. We propose the first surrogate that corrects a coarse-grained simulation via learned, grid-independent, non-local parameterizations.

**Direct numerical simulation.** Despite significant progress in simulating physics numerically it remains prohibitively expensive to repeatedly solve high-dimensional partial differential equations (PDEs) (Karniadakis et al., 2021). For example, finite difference, element, volume, and (pseudo-) spectral methods have to be re-run for every choice of initial or boundary condition, grid, or parameters (Farlow, 1993; Boyd, 2013). The issue arises if the chosen method does not have optimal runtime, i.e., does not scale linearly with the number of grid points, which renders it infeasibly expensive for calculating ensembles (Boyd, 2013). Select methods have optimal or close-to-optimal runtime, e.g., quasi-linear $O(N \log N)$, and outperform machine learning-based methods in runtime and accuracy, but their implementation often requires significant problem-specific adaptations; for example multigrid (Briggs et al., 2000) or spectral methods (Boyd, 2013). We acknowledge the existence of impressive resarch directions towards optimal and flexible non-ML solvers, such as the spectral solver called Dedalus (Burns et al., 2020), but advocate to simultaneously explore easy-to-adapt ML methods to create fast, accurate, and flexible surrogate models.

**Surrogate modeling.** Surrogate models are approximations, lightweight copies, or reduced-order models of PDE solutions, often fit to data, and used for parameter exploration or uncertainty quantificiation (Smith, 2013; Quarteroni & Rozza, 2014). Surrogate models via SVD/POD (Chatterjee, 2000), Eigendecompositions/KLE (Fukunaga & Koontz, 1970), Koopman

operators/DMD (Williams et al., 2015), take simplifying assumptions to the dynamics, e.g., linearizing the equations, which can break down in high-dimensional or nonlinear regimes (Quarteroni & Rozza, 2014). Instead, our work leverages the expressiveness of neural operators as universal approximations (Chen & Chen, 1995) to learn fast high-dimensional surrogates that are accurate in nonlinear regimes (Lütjens et al., 2021; Yuval et al., 2021; Costa Nogueira et al., 2020; Nogueira Jr. et al., 2021). **Pure ML-based** surrogate models have shown impressive sucess in approximating dynamical systems from ground-truth simulation data – for example with neural ODEs (Rackauckas et al., 2020; Chen et al., 2018; Hasani et al., 2021), GNNs (Brandstetter et al., 2022; Cachay et al., 2021a), CNNs (Stachenfeld et al., 2022), neural operators (Li et al., 2021a; Anandkumar et al., 2020; Pathak et al., 2022; Lu et al., 2021; Jiang et al., 2021), RNNs (Kani & Elsheikh, 2017; Rasp et al., 2020), GPs (Chakraborty et al., 2021), reservoir computing (Pathak et al., 2018; Nogueira Jr. et al., 2021), or transformers (Chattopadhyay et al., 2020a) – but, without incorporating physical knowlege become data-hungry and poor at generalization (Karniadakis et al., 2021; Beucler et al., 2021b).

**Physics-informed machine learning.** Two main approaches of incorporating physical knowledge into ML systems is via known symmetries (Bronstein et al., 2021) or equations (Karniadakis et al., 2021). Our approach leverages known equations for computing a coarse-grid prior; which is complementary to using known equations as soft (Raissi et al., 2019; Lee & Carlberg, 2020; Zeng et al., 2021; Wu et al., 2020; Zhang et al., 2018; Yazdani et al., 2020) or hard constraints (Greydanus et al., 2019; Lutter et al., 2019; Beucler et al., 2021a; Donti et al., 2021; Beucler et al., 2019; Jin et al., 2020) as these methods can still be used to constrain the learned parametrization. In terms of symmetry, our approach exploits translational equivariance via Fourier transformations (Li et al., 2021a), but can be extended to other frameworks that exploit in- or equivariance of PDEs (Olver, 1986) to rotational (Fuchs et al., 2020; Thomas et al., 2018), Galilean (Wu et al., 2018; Prakash et al., 2021), scale (Beucler et al., 2021b), translational (Subel et al., 2021), reflectional (Cohen & Welling, 2017) or permutational (Zhou et al., 2020) transformations.

The field of physics-informed machine learning is very broad, as reviewed most recently in (Willard et al., 2022) and (Karniadakis et al., 2021; Carleo et al., 2019; Karpatne et al., 2017). We focus on the task of learning fast and accurate surrogate models of fine-scale models when a fast and approximate coarse-grained simulation is availabe. This task differs from other interesting research areas in equation discovery or symbolic regression (Brunton et al., 2016; Long et al., 2018b; 2019; Liu et al., 2021; Qian et al., 2022), downscaling or superresolution (Xie et al., 2018; Bode et al., 2021; Kurinchi-Vendhan et al., 2021; Stengel et al., 2020; Vandal et al., 2017; Groenke et al., 2020), design space exploration or data synthesis (Chen & Ahmed; Chan & Elsheikh, 2019), controls (Bieker et al., 2020) or interpretability (Toms et al., 2020; McGraw & Barnes, 2018). Our work is complementary to data assimilation or parameter calibration (Jia et al., 2019; 2021; Karpatne et al., 2017; Zhang et al., 2019; Bonavita & Laloyaux, 2020) which fit to observational data instead of models and differs from inverse modeling and parameter estimation (Parish & Duraisamy, 2016; Hamilton et al., 2017; Yin et al., 2021; Long et al., 2018a) which usually fit parametrizations that are independent of the previous state.

**Correcting coarse-grid simulations via parametrizations.** Problems with large domains are often solved via multiscale methods (Pavliotis & Stuart, 2008). Multiscale methods simulate the dynamics on a coarse-grid and capture the effects of small-scale dynamics that occur within a grid cell via additive terms, called subgrid parametrizations, closures, or residuals (Pavliotis & Stuart, 2008; McGuffie & Henderson-Sellers, 2005). Existing subgrid parametrizations for many equations are still inaccurate (Webb et al., 2015) and ML outperformed them by learning parametrizations directly from high-resolution simulations; for example in turbulence (Duraisamy et al., 2019), climate (Gentine et al., 2018), chemistry (Hansen et al., 2013), biology (Peng et al., 2021), materials (Liu et al., 2022), or hydrology (Bennett & Nijssen, 2020). The majority of ML-based parametrizations, however, is local (Gentine et al., 2018; O'Gorman & Dwyer, 2018; Brenowitz & Bretherton, 2018; Brenowitz et al., 2020; Bretherton et al., 2022; Yuval et al., 2021; Cachay et al., 2021b; Bennett & Nijssen, 2020; Hansen et al., 2013; Liu et al., 2022; Prakash et al., 2021; Ling et al., 2016; Parish & Duraisamy, 2016; Wu et al., 2018; Rasp, 2020), i.e., the in- and output are variables of single grid points, which assumes perfect scale separation, for example, in isotropic homogeneous turbulent flows (P., 2006). However, local parametrizations are inaccurate; for example in the case of anisotropic nonhomogeneous dynamics (P., 2006; Wang et al., 2022), for correcting global error of coarse spectral discretizations (Boyd, 2013), or in large-scale climate models (Dueben & Bauer,

Figure 2: Left: **Model Architecture.** A physics-based model, $\mathcal{N}$, can quickly propagate the state, $\bar{u}_t$, at a large-scale, but will accumulate the error, $h = \overline{\mathcal{N}(u)} - \mathcal{N}\bar{u}$. A neural operator, $\mathcal{K}_\theta$, wraps the computational and implementation complexities of unmodeled fine-scale dynamics into a non-local and grid-independent term, $\hat{h}$, that iteratively corrects the large-scale model. Right: **Multiscale Lorenz96.** We demonstrate multiscale neural operator (MNO) on the multiscale Lorenz96 equation, a model for chaotic atmospheric dynamics. Image: (Rasp, 2020)

2018; Pathak et al., 2018). More recent works propose non-local parametrizations, but their formulations either rely on a fixed-resolution grid (Wang et al., 2022; Blakseth et al., 2022; Lapeyre et al., 2019; Chattopadhyay et al., 2020b), an autodifferentiable solver (Um et al., 2020; Sirignano et al., 2020; Frezat et al., 2022), or are formulated for a specific domain (Behler J, 2007). A single work proposes non-local and grid-independent parametrizations (Pathak et al., 2020), but requires the explicit representation of a high-resolution state which is computationally infeasible for large domains, such as in climate modeling. We are the first to propose grid-independent and non-local parametrizations via neural operators to create fast and accurate surrogate models of fine-scale simulations.

**Neural operators for grid-independent, non-local parametrizations.** Most current learning-based non-local parametrizations rely on FCNNs, CNNs (Lapeyre et al., 2019), or RNNs (Chattopadhyay et al., 2020b), which are mappings between finite-dimensional spaces and thus grid-dependent. In comparison, neural operators learn mappings in between infinite-dimensional function spaces (Kovachki et al., 2021) such as the Laplacian, Hessian, gradient, or Jacobian. Typically, neural operators lift the input into a grid-independent state such as Fourier (Li et al., 2021a), Eigen- (Bhattacharya et al., 2020), graph kernel (Li et al., 2020; Anandkumar et al., 2020) or other latent (Lu et al., 2021) modes and learn weights in the lifted domain. We are the first to formulate neural operators for learning parametrizations.

## 3 APPROACH

We propose *multiscale neural operator* (MNO): a surrogate model with quasilinear runtime complexity that exploits know coarse-grained simulations and learns a grid-independent, non-local parametrization. As detailed in the following MNO propagates the dynamics according to:

$$\underbrace{\frac{\partial \bar{u}}{\partial t}}_{\text{Corrected Large-scale Dyn.}} = \underbrace{\mathcal{N}(\bar{u})}_{\text{Large-scale Dyn.}} + \underbrace{\mathcal{K}_\theta(\bar{u})}_{\text{Parametrization}} \tag{1}$$

### 3.1 MULTISCALE NEURAL OPERATOR

**Partial differential equations.** We focus on partial differential equations (PDEs) that can be written as initial value problem (IVP) via the method of lines (William, 1991). The PDEs in focus have one temporal dimension, $t \in [0, T] =: D_t$, and (multiple) spatial dimensions, $x = [x_1, ..., x_d]^T \in D_x$, and can be written in the iterative, explicit, symbolic form (Farlow, 1993):

$$\frac{\partial u}{\partial t} - \mathcal{N}(u) = 0 \text{ with } t, x \in [0, T] \times D_x$$
$$u(0, x) = u^0(x), \ \mathcal{B}[u](t, x) = 0 \text{ with } x \in D_x, \tag{2}$$
$$(t, x) \in [0, T] \times \partial D_x$$

In our case, the (non-)linear operator, $\mathcal{N}$, encodes the **known** physical equations; for example a combination of Laplacian, integral, differential, etc. operators. Further, $u : D_t \times D_x \to D_u$ is the

solution to the initial values, $u^0 : D_x \to D_u$, and Dirichlet, $\mathcal{B}_D[u] = u - b_D$, or Neumann boundary conditions, $\mathcal{B}_N[u] = n^T \partial_x u - b_N$, with outward facing normal on the boundary, $n \perp \partial B$.

**Scale separation.** We transfer a concept from the rich and mathematical literature in multiscale modeling (Pavliotis & Stuart, 2008) to consider a filter kernel operator, $\mathcal{G}*$, that creates the large-scale solution, $\bar{u}(x) = u(x) + u'(x)$, where $u'$ are the small-scale deviations and $\bar{\cdot}$ denotes the filtered variable, $\bar{\phi}(x) = \mathcal{G} * \phi = \int_{D_x} G(x, x')\phi(x')dx'$. Assuming the kernel, $G$, 1) preserves constant fields, $\bar{a} = a$, 2) commutes with differentiation, $[\mathcal{G}*, \frac{\partial}{\partial s}]$ with $s = x, t$, and 3) is linear, $\overline{\phi + \psi} = \bar{\phi} + \bar{\psi}$ (P., 2006), we can rewrite (2) to:

$$\mathcal{G} * \frac{\partial u}{\partial t} = \frac{\partial \bar{u}}{\partial t} = \mathcal{G} * \mathcal{N}(u)$$
$$= \mathcal{N}(\bar{u}) + [\mathcal{G}*, \mathcal{N}](u) \tag{3}$$

where $[\mathcal{G}*, \mathcal{N}](u) = \mathcal{G} * \mathcal{N}(u) - \mathcal{N}(\mathcal{G} * u)$ is the filter subgrid parametrization, closure term, or commutation error, i.e., the error introduced through propagating the coarse-grained solution.

Approximations of the subgrid parametrization as an operator that acts on $\bar{u}$ require significant domain expertise and are derived on a problem-specific basis. In the case of isotropic homogeneous turbulence, for example, the subgrid parametrization can be approximated as the spatial derivative of the subgrid stress tensor, $[\mathcal{G}*, \mathcal{N}](\bar{u})_{\text{turbulence}} \approx \frac{\partial \tau_{ij}}{\partial x_j} = \frac{\partial \overline{u_i' u_j'}}{\partial x_j}$ (P., 2006). Many works approximate the subgrid stress tensor with physics-informed ML (Prakash et al., 2021; Ling et al., 2016; Parish & Duraisamy, 2016; Wu et al., 2018), but are domain-specific, local, or require a differentiable solver or fixed-grid. We propose a general purpose method to approximating the subgrid parametrization, independent of the grid, domain, isotropy, and underlying solver.

**Multiscale neural operator.** We aim to approximate the parametrization / filter commutation error, $[\mathcal{G}*, \mathcal{N}] \approx: h$, via learning a neural operator on high-resolution training data. Let $\mathcal{K}_\theta$ be a neural operator with the mapping:

$$[\mathcal{G}*, \mathcal{N}] \approx \mathcal{K}_\theta : \bar{U}(D_x; \mathbb{R}^{d_u}) \to H(D_x; \mathbb{R}^{d_u}) \tag{4}$$

where $\theta$ are the learned parameters and $\bar{U}, H$ are separable Banach spaces of all continuous functions taking values, $\mathbb{R}^{d_u}$, defined on the bounded, open set, $D_x \subset \mathbb{R}^{d_x}$, with norm $||f||_{\bar{U}} = ||f||_H = \max_{x \in D_x} |f(x)|$. We embed the neural operator as an autoregressive model with fixed time-discretization, $\Delta t$, such that the final *multiscale neural operator* (MNO) model is:

$$\bar{u}(t + \Delta t) = f(t, \bar{u}, \frac{\partial \bar{u}}{\partial x}, \frac{\partial^2 \bar{u}}{\partial x^2}, \dots) + \mathcal{K}_\theta(\bar{u}) \tag{5}$$

where $f(t, \bar{u}, \frac{\partial \bar{u}}{\partial x}, \frac{\partial^2 \bar{u}}{\partial x^2}) = \int_t^{t+\Delta t} \mathcal{N}(\bar{u}) d\tau$ is the known large-scale tendency, i.e. one-step solution. MNO is fit using MSE with the loss function:

$$L = \mathbb{E}_t \mathbb{E}_{\bar{u}|u(t) \sim p(t)} \left( \mathcal{L}(\mathcal{K}_\theta(\bar{u}(t)), [\mathcal{G}*, \mathcal{N}](u(t))) \right) \tag{6}$$

where the ground-truth data, $u(t) \sim p(t)$, is generated by integrating a high-resolution simulation with varying parameters, initial or boundary conditions and uniformly sampling time snippets according to the distribution $p(t)$. Similar to problems in superresolution, there exist multiple realizations of the learned commutation error, $[\mathcal{G}*, \mathcal{N}](\bar{u})$, for a given ground-truth, $[\mathcal{G}*, \mathcal{N}](u)$; using MSE will learn a smooth average and future work will explore adversarial losses (Goodfellow et al., 2014) or an intersection between neural operators and normalizing flows (Rezende & Mohamed, 2015) or diffusion-based models (Sohl-Dickstein et al., 2015) to account for the stochasticity (Wilks, 2005). During training, the model input is generated via $\bar{u}(t) = \mathcal{G} * (u(t))$ and the target via

$$h_{\text{target}} = \overline{\mathcal{N}(u)} - \mathcal{N}(\bar{u}). \tag{7}$$

During inference MNO is initialized with a large-scale state and integrates the dynamics in time via coupling the neural operator and a large-scale simulation.

Our approach does not need access to the high-resolution simulator or equations; it only requires a precomputed high-resolution dataset, which are increasingly available (Hersbach et al., 2020; Burns et al., 2022), and allows the user to incorporate existing easy-to-access solvers of large-scale equations. There is no requirement for the large-scale solver to be autodifferentiable which significantly simplifies the implementation for large-scale models, such as in climate. If desired, our loss function can easily be augmented with a physics-informed loss (Raissi et al., 2019) on the large-scale dynamics or parametrization term.

**Choice of neural operator.**    Our formulation is general enough to allow the use of many operators, such as Fourier (Li et al., 2021a), PCA-based (Bhattacharya et al., 2020), low-rank (Khoo & Ying, 2019), Graph (Li et al., 2020) operators, or DeepOnet (Wang et al., 2021; Lu et al., 2021). Because DeepONet (Lu et al., 2021) focuses on interpolation and assumes fixed-grid sensor data, we decided to modify Fourier Neural Operator (FNO) (Li et al., 2021a) for our purpose. FNO is a universal approximator of nonlinear operators (Kovachki et al., 2021; Chen & Chen, 1995), grid-independent and can be formulated as autoregressive model (Li et al., 2021a). As there exists significant knowledge on symmetries and conservation properties of the commutation error (P., 2006), MNO's explicit formulation increases interpretability and ease of incorporating symmetries and constraints. With FNO, we exploit approximate translational symmetries in the data and leave novel opportunities for neural operators that exploit the full range of known equi- and invariances of the subgrid parametrization term, such as Galilean invariance (Prakash et al., 2021), for future work.

## 3.2    Illustration of MNO via multiscale Lorenz96

We illustrate the idea of MNO on a canonical model of atmospheric dynamics, the multiscale Lorenz96 equation Lorenz (2006); Thornes et al. (2017). This PDE is multiscale, chaotic, time-continuous, space-discretized, 2D (space+time), nonlinear, displayed in Fig. 2-right and detailed in Appendix A.3. Most importantly, the large- and small-scale solutions, $X_k \in \mathbb{R}, Y_{j,k} \in \mathbb{R} \ \forall \ j \in \{0, ..., J\}, k \in \{0, ..., K\}$, demonstrate the *curse of dimensionality*: the number of the small-scale states grows exponentially with scale and explicit modeling becomes computationally expensive, for example, quadratic for two-scales: $O(N^2) = O(JK)$. The PDE writes:

$$\frac{\partial X_k}{\partial t} = \underbrace{X_{k-1}(X_{k+1}-X_{k-2})-X_k+F}_{\text{Large-scale Dyn.: } \frac{\partial \bar{X}_k}{\partial t}} \underbrace{-\frac{h_s c}{b}\sum_{j=0}^{J-1} Y_{j,k}(X_k)}_{\text{Parametrization: } h},$$

$$\frac{\partial Y_{j,k}}{\partial t} = -cbY_{j+1,k}(Y_{j+2,k}-Y_{j-1,k})-cY_{j,k}+\frac{h_s c}{b}X_k. \tag{8}$$

where $F$ is the forcing, $h_s$ the coupling strength, $b$ the relative magnitude of scales, and $c$ the evolution speed. With the multiscale framework from Section 3.1, we define:

$$u(x) = [X_0, Y_{0,0}, Y_{1,0}, ..., Y_{J,0}, X_1, Y_{0,1}, ...$$
$$, X_K, ..., Y_{J,K}]_x \ \forall x \in D_x = \{0, ..., K(J+1)\}$$

$$\mathcal{N}(u)(x) = \begin{cases} \frac{\partial X_k}{\partial t} & \text{if } x=k(J+1) \ \forall k \in \{0, \dots, K\} \\ \frac{\partial Y_{j,k}}{\partial t} & \text{otherwise,} \end{cases}$$

$$G(x, x') = \begin{cases} 1 \text{ if } x' = k(J+1) \ \forall k \in \{0, \dots, K\} \\ 0 \text{ otherwise,} \end{cases}$$

with the solution, $u$, operator, $\mathcal{N}$, and kernel, $G$.

MNO learns the parametrization term via a neural operator, $\mathcal{K}_\theta = \hat{h} \approx h$, and then models:

$$\frac{\partial \hat{X}_k}{\partial t} = \frac{\partial \overline{\hat{X}}_k}{\partial t} + \mathcal{K}_\theta(\hat{X}_{0:K})(k) \tag{9}$$

where the known large-scale dynamics are approximated with $\frac{\partial \overline{\hat{X}}_k}{\partial t} \approx \frac{\partial \overline{X}_k}{\partial t}$ and ground-truth parametrization is $h(x) = \{-\frac{h_s c}{b}\sum_{j=0}^{J-1} Y_{j,k}(X_k) \text{ if } x = k(J+1) \ \forall k \in \{0, \dots, K\} \text{ and } 0 \text{ otherwise}\}$. See Appendix A.4 for all terms.

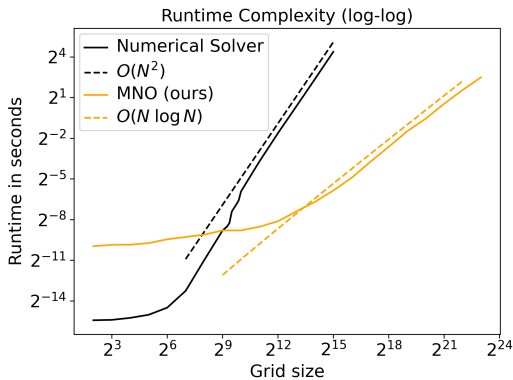

Figure 3: **MNO is faster than direct numerical simulation.** Our proposed multiscale neural operator (orange) can propagate multiscale PDE dynamics in quasilinear complexity, $O(N \log N)$. For a grid with $K = 2^{15}$, MNO is $\sim$ 1000-times faster than direct numerical simulation (black) which has quadratic complexity, $O(N^2)$

The parametrization, $\mathcal{K}_\theta$, accepts inputs that are sampled anywhere inside the spatial domain, which differs from previous local (Rasp, 2020) or grid-dependent (Chattopadhyay et al., 2020b) Lorenz96 parametrizations.

We create the ground-truth data via randomly sampled initial conditions, periodic boundary conditions, and integrating the coupled equation with a 4th-order Runge-Kutta solver. After a Lyapunov timescale the state is independent of initial conditions and we extract $4K$ snippets with $T/\Delta t = 400$steps length, corresponding to 10 Earth days, for 1-step training. During test the model is run autoregressively on 1K samples from a different initial condition, as detailed in Appendix A.3.

## 4 RESULTS

Our results demonstrate that multiscale neural operator (MNO) is faster than direct numerical simulation, generates stable solutions, and is more accurate than current parametrizations. We now proceed to discussing each of these in more detail.

### 4.1 RUNTIME COMPLEXITY: MNO IS FASTER THAN TRADITIONAL PDE SOLVERS

MNO (orange in Fig. 3) has quasilinear, $O(N \log N)$, runtime complexity in the number of large-scale grid points, $N=K$, in the multiscale Lorenz96 equation. The runtime is dominated by a lifting operation, here a fast Fourier transform (FFT), which is necessary to learn spatial correlations in a grid-independent space. In comparison, the direct numerical simulation (black) has quadratic runtime complexity, $O(N^2)$, because of the explicit representation of $N^2=JK$ small-scale states. Both models are linear in time, $O(T)$. Local parametrizations can achieve optimal runtime, $O(N)$, but it is an open question if there exists a decomposition that replaces FFT to yield an optimal, non-local, grid-independent model.

We ran MNO up to a resolution of $K = 2^{24}$, which would equal $75cm/px$ in a global 1D (space) climate model and only took $\approx 2s$ on a single CPU. MNO is three orders of magnitude (1000-times) faster than DNS, at a resolution of $K = 2^{15}$ or $200m/px$. For 2D or 3D simulations the gains of using MNO vs. DNS are even higher with $O(N^2 \log N)$ vs. $O(N^4)$ and $O(N^3 \log N)$ vs. $O(N^6)$, respectively (Khairoutdinov et al., 2005).

The runtimes have been calculated by choosing the best of 1-100k runs depending on grid size on a single-threaded Intel Xeon Gold 6248 CPU@2.50GHz with 164Gb RAM. We time a one step update which, for DNS, is the calculation of (8) and for MNO the calculation of (9), i.e., the sum of a large-scale step and a pass through the neural operator.

In Fig. 3, the runtime of MNO and DNS plateaus at low-resolution ($K < 2^9$), because runtime measurement is dominated by grid-independent operations. DNS plateaus at a lower runtime, be-

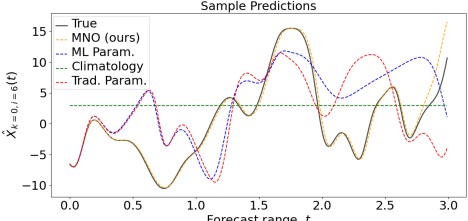

| Method | RMSE |
|---|---|
| Climatology | 6.902 |
| Traditional parametrizations | 2.326 |
| ML-based parametrization | |
| (Rasp et al., 2018) | 2.053 |
| **MNO (ours)** | 0.5067 |

Figure 4: **Left: MNO is more accurate than traditional parametrizations.** A sample plot shows, that our proposed multiscale neural operator (yellow/orange-dotted) can accurately forecast the large-scale physics (black-solid), $X_{k=0}(t)$. In comparison, ML-based blue-dotted) and traditional (red-dotted) parametrizations quickly start to diverge. Note that the system is chaotic and small deviations are rapidly amplified; even inserting the exact parametrizations in float32 instead of float64 quickly diverges. **Right: Accuracy.** MNO is more accurate than traditional parametrizations as measured by the root mean-square error (RMSE).

cause MNO contains several fixed-cost matrix transformations. The runtime of DNS has a slight discontinuity at $K \approx 2^9$ due to extending from cache to RAM memory. We focus on a runtime comparison, but MNO also has significant savings in memory: representing the state at $K = 2^{17}$ in double precision occupies $64$GB RAM for DNS and $0.5$MB for MNO.

## 4.2 MNO IS MORE ACCURATE THAN TRADITIONAL PARAMETRIZATIONS

Figure 4-left shows a forecasted trajectory of a sample at the left boundary, $k = 0$, where MNO (orange-dotted) accurately forecasts the large-scale dynamics, $X_0(t)$, (black-solid) while current ML-based (blue-dotted) (Gentine et al., 2018) and traditional parametrizations (red-dotted) quickly diverge. The quantitive comparison of RMSE and a mean/std plot Fig. 4 over $1K$ samples and $200$steps or $10$days ($\Delta t = 0.005 = 36$min) confirms that MNO is the most accurate in comparison to ML-based parametrizations, traditional parametrizations, and a mean forecast (climatology). Note, the difficulty of the task: when forecasting *chaotic* dynamics even numerical errors rapidly amplify (P., 2006).

**ML-based parametrizations** is a state-of-the-art (SoA) model in learning parametrizations and trains a ResNet to forecast a local, grid-independent parametrization, $h_k = \text{NN}(X_k)$, similar to (Gentine et al., 2018). The **traditional parametrizations** (trad. param.) are often used in practice and use linear regression to learn a local, grid-independent parametrization (McGuffie & Henderson-Sellers, 2005). It was suggested that multiscale Lorenz96 is too easy as a test-case for comparing offline models because traditional parametrizations already perform well (Rasp, 2019), but the significant difference between MNO and Trad. Params. shows that online evaluation is still interesting.

The **climatology** forecasts the mean of the training dataset, $X_k(t) = 1/T \sum_{t=0}^{T} 1/N \sum_{i=0}^{N} X_{k,i}(t)$. The full list of hyperparameters and model parameters can be found in Appendix A.5.2. For fairness, we only compare against grid-independent methods that do not require an autodifferentiable solver. Models with soft or hard constraints, e.g., PINNs (Raissi et al., 2019) or DC3 (Donti et al., 2021), are complementary to MNO. Further, note that our implementation of MNO uses an a priori loss function and could likely be improved by implementing an a posteriori loss functions, i.e., a loss functions that propagates the loss over multiple time steps similar to (Frezat et al., 2022) which requires an autodifferentiable solver or (Brandstetter et al., 2022) which does not.

## 4.3 MNO IS STABLE

Figure 5 shows that predicting large-scale dynamics with MNO is stable. We first plot a randomly selected sample of the first large-scale state, $X_{k=0}(t)$ (left-black), to illustrate that the prediction is bounded. The MNO prediction (left-yellow) follows the ground-truth up to an approximate horizon of, $t = 1.8$ or 9 days, then diverges from the ground-truth solution, but stays within the bounds of the ground-truth prediction and does not diverge to infinity. The RMSE over time in Figure 5 shows that MNO (yellow) is approximately more accurate than current ML-based (blue) and traditional (red) parametrizations for $\approx 100\%$-longer time, measuring the time to intersect with climatology. Despite the difficulty in predicting chaotic dynamics, the RMSE of MNO reaches a plateau, which is slightly above the optimal plateau given by the climatology (black).

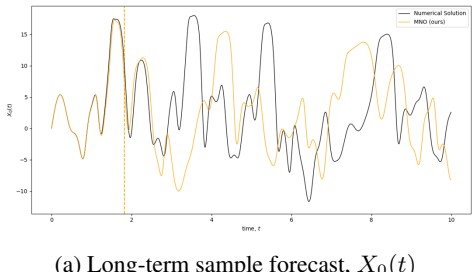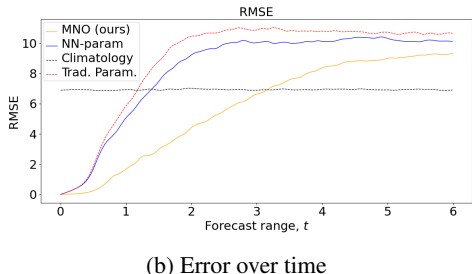

(a) Long-term sample forecast, $X_0(t)$        (b) Error over time

Figure 5: **MNO is stable.** MNO can propagate a sample state, $X_{k=0}(t)$, over a long time horizon without diverging to infinity (left). The right plot shows that the RMSE of MNO plateaus for long-term forecasts, further confirming stability. Further, MNO (yellow) maintains accuracy longer than ML-based parametrizations (blue) and a climatology (black).

The RMSE over time is calculated as:

$$\text{RMSE}(t) = \frac{1}{K} \sum_{k=0}^{K} \sqrt{\left(\frac{1}{N} \sum_{i=0}^{N} (\hat{X}_{k,i}(t) - X_{k,i}(t))^2\right)}. \tag{10}$$

## 5 LIMITATIONS AND FUTURE WORK

We demonstrated the accuracy, speed, and stability of MNO on the chaotic multiscale Lorenz96 equation. Future work, can extend MNO towards higher-dimensional or time-irregular systems and further integrate symmetries or constraints:

The results show promise to extend MNO to higher-dimensional, chaotic, multiscale, multiphysics problems. We demonstrate the first steps towards quasi-geostrophic turbulence and Rayleigh-Benard convection in Appendix A.1 and aim towards integration in large-scale global atmospheric models, e.g., as approximation of cloud processes (Wang et al., 2022; Palmer et al., 2019). Reducing the cost of climate models could dramatically improve uncertainties (Lütjens et al., 2021) or decision-exploration (Rooney-Varga et al., 2020).

MNO is grid-independent in space but not in time which could be alleviated via integrations with Neural ODEs (Chen et al., 2018). MNO is a myopic model which might suffice for chaotic dynamics (Li et al., 2021b), but could be combined with LSTMs (Mohan et al., 2019) or reservoir computing (Pathak et al., 2018) to contain a memory. Further, we leveraged global Fourier decompositions to exploit grid-independent periodic spatial correlations, but future work could also capture local discontinuities, e.g., along coastlines (Jiang et al., 2021) with multiwavelets (Gupta et al., 2021), or incorporate non-periodic boundaries via Chebyshev polynomials.

Lastly, MNO can be combined with Geometric deep learning, PINNs, or hard constraint models. This avenue of research is particularly exciting with MNO as there exist many known symmetries for various paramtrization terms (Prakash et al., 2021).

## 6 CONCLUSION

We proposed a hybrid physics-ML surrogate of multiscale PDEs that is quasilinear, accurate, and stable. The surrogate limits learning to the influence of fine- onto large-scale dynamics and is the first to use neural operators for a grid-independent, non-local corrective term of large-scale simulations. We demonstrated that multiscale neural operator (MNO) is faster than direct numerical simulation ($O(N \log N)$ vs. $O(N^2)$), and more accurate ($\approx 100\%$ longer prediction horizon) than state-of-the-art parametrizations on the chaotic, multiscale equations multiscale Lorenz96. With the dramatic reduction in runtime MNO could enable rapid parameter exploration and robust uncertainty quantification in complex climate models.

## 7 ETHICS STATEMENT

Climate change is a defining challenge of our time and environmental disasters will become more frequent: from storms, floods, wildfires and heat waves to biodiversity loss and air pollution (IPCC, 2018). The impacts of these environmental disasters will likely be unjustly distributed: island states, minority populations, and the Global South are already facing the most severe consequences of climate change, while the Global North is responsible for the most emissions since the industrial revolution (Althor et al., 2016). Decision-makers require more accurate, accessible, and local tools to understand and limit the economic and human impact of a changing climate (Palmer et al., 2019). We propose multiscale neural operator (MNO) to improve the parametrizations in climate models, thus leading to more accurate predictions. Related techniques to MNO, specifically neural operator-based surrogate models, could help reduce computational complexity of large-scale weather and climate models. The reduced computational complexity would make them more accessible to low-resource countries or allow for higher resolution predictions. Unfortunately, discoveries for faster differential equations solvers can and likely will be leveraged in ethically questionable fields, such as missile development or oil discovery. We acknowledge the possible negative impacts and hope that our targeted discussion and application to equations from climate modeling can steer the our work towards a positive impact.

## 8 REPRODUCIBILITY STATEMENT

The code will be submitted in a comment directed to the reviewers and area chairs with a link to an anonymous repository once the discussion forum is opened. The code will be made publicly available upon acceptance along with an open-source license, data, and instructions to reproduce the main results. A list of hyperparameters per model is detailed in Appendix A.6.2, data splits are explained in results; all simulation details in Appendix A.4; and background of the neural operator model in Appendix A.3. Conducting the study from ideation to publication used a total of approximately 10K CPU hours on an internal cluster.

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

# A APPENDIX

## A.1 QUASI-GEOSTROPHIC TURBULENCE

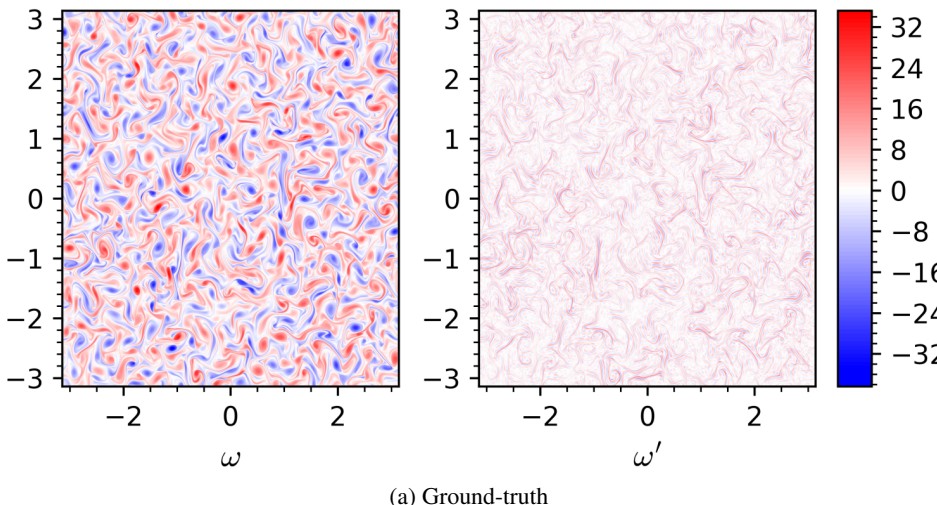

(a) Ground-truth

Figure 6: **Quasi-Geostrophic Turbulence (QGT).** We demonstrate the first steps to extend MNO to 2D QGT. This plot shows our ground-truth training data with the direct numerical simulation, $\omega$, and subgrid parametrization, $\omega'$.

We are planning to demonstrate multiscale neural operator a high-dimensional system, specifically the one-layer quasi-geostrophic (QG) turbulence depicted in Fig. 6. QG turbulence is a derivative of the Navier-Stokes equations and a good model for atmospheric turbulence at the equator. The equations are derived from the incompressible, i.e., $\nabla \cdot \boldsymbol{u} = 0$, Navier-Stokes equations by 1) taking the curl of velocity field, $w = \nabla \times \boldsymbol{u}$, and 2) assuming the beta-plane approximation, $f = f_0 + \beta y$, and hydrostatic, $\frac{\partial p}{\partial z} = -\rho g$, and geostrophic, $fv = \frac{1}{\rho}\frac{\partial p}{\partial x}; \ fu = -\frac{1}{\rho}\frac{\partial p}{\partial y}$, balances (Majda & Wang, 2006). The resulting equations, called quasi-geostrophic turbulence, are given by:

$$\partial_t \omega + J(\psi, \omega) = \nu \nabla^2 \omega - \mu\omega - \beta\partial_x\psi + F$$
$$\omega = \nabla^2 \psi \tag{11}$$

where $\omega$ is the vorticity, $\boldsymbol{u} = [u, v]^T = [-\partial_y\psi, \partial_x\psi]^T$ is the velocity vector, $\psi$ is the streamfunction, $J(\psi, \omega) = \partial_x\psi\partial_y\omega - \partial_y\psi\partial_x\omega$ is the nonlinear Jacobian operator. Further, the parameters are the turbulent viscosity, $\nu$, linear drag coefficient, $\mu$, Rossby parameter, $\beta$, and source term, $F$. Vorticity can be computed with $\omega = \hat{\boldsymbol{z}} \cdot \nabla \times \boldsymbol{u} = \partial_x v - \partial_y u$

Filtering the equation with a kernel results in the parametrized large-scale equation given by:

$$\partial_t\bar{\omega} + J(\bar{\psi}, \bar{\omega}) = \nu\nabla^2\bar{\omega} - \mu\bar{\omega} - \beta\partial_x\bar{\psi} + \bar{F} + \underbrace{J(\bar{\psi}, \bar{\omega}) - \overline{J(\psi, \omega)}}_{\text{Parametrization: } h(\psi, \omega)} \tag{12}$$

We then aim to approximate the subgrid-scale (SGS) parametrization, $\mathcal{K}_\theta \approx h$, with the neural operator, such that the final model is:

$$\frac{\partial\omega(x, y)}{\partial t} = \frac{\partial\bar{\omega}(x, y)}{\partial t} + \mathcal{K}_\theta(\psi, \omega)(x, y)_\omega$$
$$\frac{\partial\psi(x, y)}{\partial t} = \frac{\partial\bar{\psi}(x, y)}{\partial t} + \mathcal{K}_\theta(\psi, \omega)(x, y)_\psi \tag{13}$$

The QG turbulence equation is solved with a pseudospectral solver in space and RK4 explicit time integration. We choose the parameters, $N_x = N_y = 512$, $\Delta t = 480$s, $\mu = 1.25 \times 10^{-8}\text{s}^{-1}$, $\nu = 352\text{m}^2/\text{s}$, $\beta = 0$, and Reynolds number, $\text{Re} = 22 \times 10^4$. The variables are non-dimensionalized with $T_d = 1.2 \times 10^6 s$, i.e., $\Delta t_{\text{solver}} = \Delta t/T_d$ and $L_d = 504 \times 10^4/\pi m$, i.e., $\Delta x_{\text{solver}} = 2\pi/N_x$. The

reduced system is run with scale $\delta = 4$, such that $\bar{N}_x = \bar{N}_y = 128$. The forcing initiates turbulent mixing and simulates slowly varying wind stress according to the solution of, $F = C_f(t)[\cos(4y + \pi\sin(1.4t)) - \cos(4x + \pi\sin(1.5t))]$, and $0.5\|F\|_2 = 3$ with enstrophy injection rate, $C_F(t)$. To generate turbulent chaotic dynamics that are decoupled from the initial state, the simulation is initialized with some large-scale Fourier states and warmed up for 1300days. After warm-up we generate 18000 iterations ($\delta18000$ at the fine-scale) from which we independently sample training and validation snippets.

## A.2 RAYLEIGH-BÉNARD CONVECTION

The proposed multiscale neural operator can also be leveraged for systems without explicit multiscale formulation. We demonstrate this by formulating the MNO equations for Rayleigh-Bénard Convection equations, as displayed in Fig. 7.

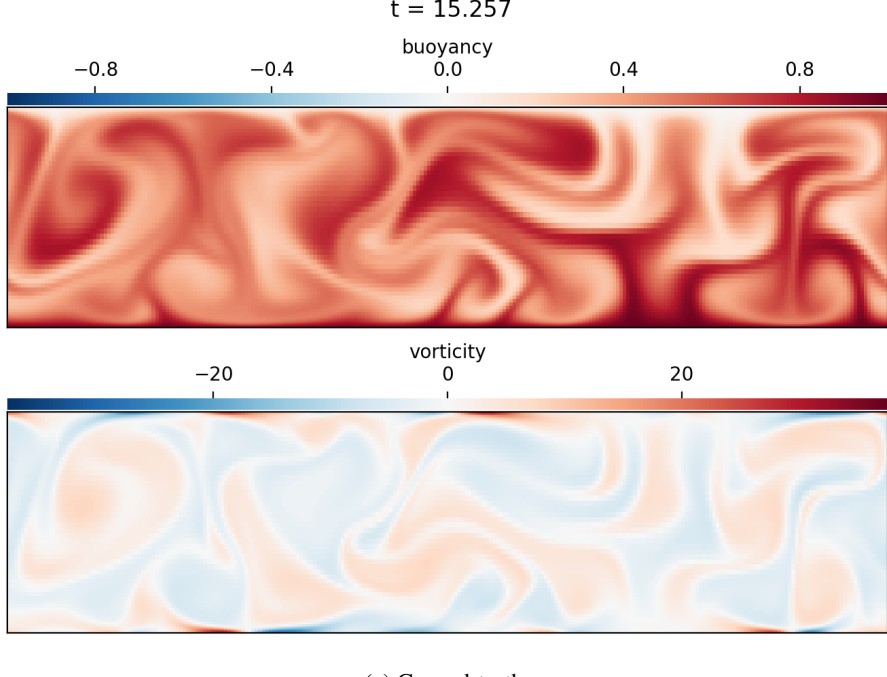

(a) Ground-truth

Figure 7: **Rayleigh-Bénard Convection.** We depicted a sample plot for ground-truth training data of the 2D RBC.

### A.2.1 DETAILS AND INTERPRETATION

Rayleigh-Bénard Convection (RBC) is a challenging set of equations for turbulent, chaotic, and convection-dominated flows. The equation finds applications in fluid dynamics, atmospheric dynamics, radiation, phase changes, magnetic fields, and more (Pandey et al., 2018).

So far, we have generated a ground-truth dataset that we implemented with the 2D turbulent Rayleigh-Bénard Convection equations with Dedalus spectral solver (Burns et al., 2020) similar to (Pandey et al., 2018):

$$\frac{\partial u}{\partial t} + u \cdot \nabla u = \sqrt{\frac{\text{Pr}}{\text{Ra}}}\nabla^2 u - \nabla p + b$$
$$\frac{\partial T}{\partial t} + u \cdot \nabla T = \frac{1}{\sqrt{RaPr}}\nabla^2 T \qquad (14)$$
$$\nabla \cdot u = 0$$

with temperature/buoyancy, $T$, Rayleigh number, $\mathrm{Ra} = g\alpha\Delta T H^3/(\nu\kappa)$, thermal expansion co-efficient, $\alpha$, Prandtl number, $\mathrm{Pr} = \nu/\kappa$, momentum diffusivity or kinematic viscosity, $\nu$, thermal diffusivity, $\kappa = \frac{1}{\sqrt{\mathrm{RaPr}}}$, acceleration due to gravity, $g$, temperature difference, $\Delta T$, unit vector, $e$, pressure, $p$, Nusselt number, $\mathrm{Nu} = \sqrt{\frac{\mathrm{Pr}}{\mathrm{Ra}}}$, Reynolds number, $Re = \sqrt{\langle\nabla^2 u\rangle_{V,t}\frac{\mathrm{Ra}}{\mathrm{Pr}}}$, and full volume-time average, $\langle\cdot\rangle_{V,t}$, cell length, $L_x$. The equations have been non-dimensionalized with the free-fall velocity, $U_f = \sqrt{g\alpha\Delta H}$, and cell height, $H$. In the horizontal direction, $x$, we use periodic boundary conditions and in the vertical direction, $z$, we use no-slip boundary conditions for the velocity, $u(z=0) = u(z=L_z) = 0$, and fixed temperatures, $T(z=0) = L_z$, $T(z=L_z) = 0$. The inital conditions are sampled randomly, $b(z,t=0) = L_z + z + z(L_z - z)\omega$, with $\omega \sim \mathcal{N}(0, 1\times 10^{-3})$.

We chose: $\mathrm{Ra} = 2\times 10^6$, $\mathrm{Pr} = 1$, $L_x = 4$, $H = 1$.

## A.3 FOURIER NEURAL OPERATOR

Our neural operator for learning subgrid parametrizations is based on Fourier neural operators (Li et al., 2021a). Intuitively, the neural operator learns a parameter-to-solution mapping by learning a global convolution kernel. In detail, it learns the operator to transforms the current large-scale state, $\underline{X}(x_{0:K}, t) \in \mathbb{R}^{K\times d_X}$ to the subgrid parametrization, $\underline{\hat{f}}_x(x_{0:K}, t) := \underline{X}_{0:K} \in \mathbb{R}^{K\times d_X}$ with number of grid points, $K$, and input dimensionality, $d_X$, according to the following equations:

$$
\begin{aligned}
\underline{v}_0 &= \underline{X}_{0:K}P^T + 1^{K\times 1}b_P \\
\underline{v}_{i+1} &= \sigma\left(\underline{v}_i W^T + \int_{D_x}\kappa_\phi(x,x')v_i(x')dx'\right) \\
&\approx \sigma\left(\underline{v}_i W^T + 1^{n_v\times 1}b_W + \mathcal{F}^{-1}(R_\phi\cdot\mathcal{F}\underline{v}_i)\right) \\
\hat{f}_{x,0:K} &= \underline{v}_{n_d}Q^T + 1^{K\times 1}b_Q
\end{aligned}
\tag{15}
$$

First, MNO lifts the input via a linear transform with matrix, $P \in \mathbb{R}^{n_v\times d_x}$, bias, $b_P \in \mathbb{R}^{1\times n_v}$, vector of ones, $1^{K\times 1}$, and number of channels, $n_v$. The linear transform is local in space, i.e., the same transform is applied to each grid point.

Second, multiple nonlinear "Fourier layers" are applied to the encoded/lifted state. The encoded/lifted state's, $\underline{v}_i \in \mathbb{R}^{K\times n_v}$, spatial dimension is transformed into the Fourier domain via a fast Fourier transform. We implement the FFT as a multiplication with the pre-built forward and inverse Type-I DST matrix, $\mathcal{F} \in \mathbb{C}^{k_{\max}\times K}$ and $\mathcal{F}^{-1} \in \mathbb{C}^{K\times k_{\max}}$, respectively, returning the vector, $\mathcal{F}\underline{v}_i \in \mathbb{C}^{k_{\max}\times n_v}$.

The dynamics are learned via convoluting the encoded state with a weight matrix. In Fourier space, convolution is a multiplication, hence each frequency is multiplied with a complex weight matrix across the channels, such that $R \in \mathbb{C}^{k_{\max}\times n_v\times n_v}$. In parallel to the convolution with $R$, the encoded state is multiplied with the linear transform, $W \in \mathbb{R}^{n_v\times n_v}$, and bias, $b_W \in \mathbb{R}^{1\times n_v}$. From a representation learning-perspective, the Fourier decomposition as a fast and interpretable feature extraction method that extracts smooth, periodic, and global features. The linear transform can be interpreted as residual term concisely capturing nonlinear residuals.

So far, we have only applied linear transformations. To introduce nonlinearities, we apply a nonlinear activation function, $\sigma$, at the end of each Fourier layer. While the non-smoothness of the activation function ReLu, $\sigma(z) = \max(0, z)$, could introduce unwanted discontinuities in the solution, we choose it resulted in more accurate models than smoother activation functions such as tanh or sigmoid.

Finally, the transformed state, $v_{n_d}$, is projected back onto solution space via another linear transform, $Q \in \mathbb{R}^{d_X\times n_v}$, and bias, $b_Q$.

The values of all trainable parameters, $P, R, W, Q, b_*$, are found by using a nonlinear optimization algorithm, such as stochastic gradient descent or, here, Adam Kingma & Ba (2015). We have used MSE between the predicted, $\hat{f}_x$, and ground-truth, $f_x$, subgrid parametrizations as loss. The neural operator is implemented in pytorch, but does not require an autodifferentiable PDE solver to generate training data. During implementation, we used the DFT which assumes a uniformly spaced grids,

but can be exchanged with non-uniform DFTs (NUDFT) to transform non-uniform grids (Dutt & Rokhlin, 1993).

## A.4 MULTISCALE LORENZ96

### A.4.1 DETAILS AND INTERPRETATION

The equation contains $K$ variables, $X_k \in \mathbb{R}$, and $JK$ small-scale variables, $Y_{j,k} \in \mathbb{R}$ that represent large-scale or small-scale atmospheric dynamics such as the movement of storms or formation of clouds, respectively. At every time-step each large-scale variable, $X_k$, influences and is influenced by $J$ small-scale variables, $Y_{0:J,k}$. The coupling could be interpreted as $X_k$ causing static instability and $Y_{j,k}$ causing drag from turbulence or latent heat fluxes from cloud formation. The indices $k, j$ are both interpreted as latitude, while $k \in \{0, ..., K{-}1\}$ indexes boxes of latitude and $j \in \{0, ..., J{-}1\}$ indexes elements inside the box. Illustrated on a 1D Earth with a circumference of $360°$ that is discretized with $K = 36, J = 10$, one a spatial step in $k, j$ would equal $10°, 1°$, respectively Lorenz (2006); we choose $K = J = 4$. A time step with $\Delta t = 0.005$ would equal 36 minutes Lorenz (2006).

We choose a large forcing, $F > 10$, for which the equation becomes chaotic. The last terms in each equation capture the interaction between small- and large-scale, $f_{x,k} = -\frac{hc}{b} \sum_{j=0}^{J} Y_{j,k}(X_k), f_y$. The scale interaction is defined by the parameters where $h = 0.5$ is the coupling strength between spatial scales (with no coupling if $h$ would be zero), $b = 10$ is the relative magnitude, and $c = 8$ the evolution speed of $X - Y$. The linear, $-X_k$, and quadratic terms, $X_*^2$, model dissipative and advective (e.g., moving) dynamics, respectively.

The equation assumes perfect "scale separation" which means that small-scale variables of different grid boxes, $k$, are independent of each other at a given timestep, $Y_{j_1,k_2}(t) \perp Y_{j_2,k_1}(t) \, \forall t, j_1, j_2, k_1 \neq k_2$. The separation of small- and large-scale variables can be along the same or different domain and the discretized variables would then be $y \in [0, \Delta x]$ or $y \in [y_0, y_{\text{end}}]$, respectively. The equation wraps around the full large- or small-scale domain by using periodic boundaries, $X_{-k}{:=}X_{K-k}$, $X_{K+k}{:=}X_k$, $Y_{-j,k}{:=}Y_{J-j,k}$, $Y_{J+j,k}{:=}Y_{j,k}$. Note that having periodic boundary conditions in the small-scale domanin allows for superparametrization, i.e., independent simulation of the small-scale dynamics Campin et al. (2011) and differs from the three-tier Lorenz96 where variables at the borders of the small-scale domain depend on small-scale variables of the neighbouring k Thornes et al. (2017).

### A.4.2 SIMULATION

The initial conditions are sampled uniformly from a set of integers, $X(t_0) \sim U(-5, -4, ..., 5, 6)$, as a mean-zero unit-variance Gaussian $Y(t_0) \sim \mathcal{N}(0, 1)$, and lower scale Gaussian $Z(t_0) \sim 0.05\mathcal{N}(0, 1)$. The train and test set contains 4k and 1k samples, respectively. Each sample is $T = 1$ model time unit (MTU) or 200 (=$T/\Delta t$) time-steps long, which corresponds to 5 Earth days (= $T/\Delta t * 36$min with $\Delta t = 0.005$) Lorenz (2006). Hence, our results test the generalization towards different initial conditions, but not robustness to extrapolation or different choices of parameters, $c, b, h, F$. The sampling starts after $T = 10$. warmup time. The dataset uses double precision.

We solve the equation by fourth order Runge-Kutta in time with step size $\Delta t = 0.005$, similar to (Lorenz & Emanuel, 1998). For a PDE that is discretized with fixed time step, $\Delta t$, the ground-truth train and test data, $h_{x,0:K}(t)$, is constructed by integrating the coupled large- and small-scale dynamics.

Note, that the neural operator only takes in the current state of the large-scale dynamics. Hence, , i.e., it uses the full large-scale spatial domain as input, which exploits spatial correlations and learns parametrizations that are independent of the large-scale spatial discretization.

Our method can be queried for infinite time-steps into the future as it does not use time as input.

We are incorporating the prior knowledge from physics by calculating the large-scale dynamics, $dX_{LS,0:K}$. Note that the small-scale physics do not need to be known. Hence, MNO could be applied to any fixed time-step dataset for which an approximate model is known.

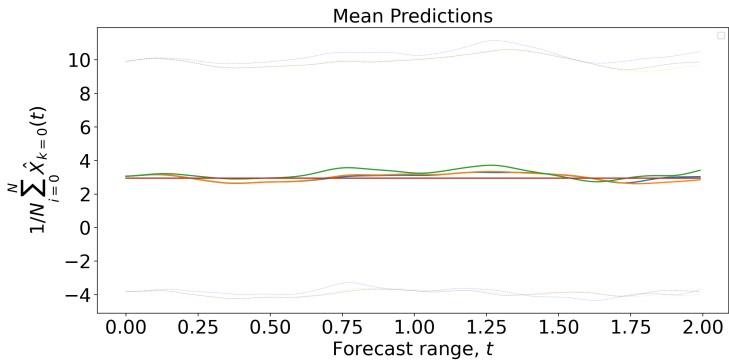

Figure 8: **Mean accuracy.** MNO (orange) forecasts the mean (solid) of the ground-truth DNS (blue) more accurately in comparison to ML-based parametrizations (green) and climatology (red). The standard deviations is plotted as dotted lines.

.

### A.5 Appendix to Illustration of MNO via multiscale Lorenz96

The other large-scale (LS) and fine-scale (FS) terms are

$$
\text{filtered FS dynamics}, \overline{\mathcal{N}(u)}(x) = \begin{cases} \frac{\partial X_k}{\partial t} & \text{if } x = k(J+1) \ \forall k \in \{0, \ldots, K\} \\ 0 & \text{otherwise} \end{cases}
$$

$$
\text{LS dynamics}, \mathcal{N}(\bar{u})(x) = \begin{cases} \frac{\partial \bar{X}_k}{\partial t} & \text{if } x = k(J+1) \ \forall k \in \{0, \ldots, K\} \\ 0 & \text{otherwise} \end{cases} \tag{16}
$$

$$
\text{with abbreviation}, \frac{\partial \bar{X}_k}{\partial t} := X_{k-1}(X_{k+1} - X_{k-2}) - X_k + F
$$

$$
\text{LS state}, \bar{u}(x) = \mathcal{G} * u(x) = [X_0, 0, ..., 0, X_1, 0, ..., X_K]
$$

### A.6 Appendix to Results

#### A.6.1 Accuracy

Figure 8 shows that the predicted mean and standard deviation of MNO (orange) closely follows the ground-truth (blue). The ML-based parametrization (green) follows the ground-truth only for a few time steps (until $\sim t = 0.125$). The climatology (red) depicts the average prediction in the training dataset.

#### A.6.2 Model configuration

**Multiscale Lorenz96: MNO**  As hyperparameters we chose the number of channels, $n_v = 64$, number of retained modes, $k_{\max} = 3$, number of Fourier layers, $n_d = 3$, and no batch norm layer. The time-series modeling task uses a history of only one time step to learn chaotic dynamics (Li et al., 2021b). We are using ADAM optimizer with learning rate, $\lambda = 0.001$, step size, 20, number of epochs, $n_e = 2$, and an exponential learning rate scheduler with gamma, $\gamma = 0.9$ (Kingma & Ba, 2015). Training took $1:50$min on a single core Intel i7-7500U CPU@2.70GHz.

**Multiscale Lorenz96: ML-based parametrization**  The ML-based parametrizations uses a ResNet with $n_{\text{layers}} = 2$ residual layers that contain a fully connected network with $n_{\text{units}} = 32$ units. The model is optimized with Adam (Kingma & Ba, 2015) with learning rate $0.01$, $\beta = (0.9, 0.999)$, $\epsilon = 1 * 10^{-8}$, trained for $20 n_{\text{epochs}} = 20$.

**Multiscale Lorenz96: Traditional parametrization**  The traditional parametrization uses least-squares to find the best linear fit. The weight matrix is computed with $A = (X^T X)^{-1} X^T Y$, where $X$ and $Y$ are the concatenation of input large-scale features and target parametrizations, respectively. Inference is conducted with $\hat{y} = Ax$.

### A.7 Neural networks vs. neural operators

Most work in physics-informed machine learning relies on fully-connected neural networks (FC-NNs) or convolutional neural networks (Karniadakis et al., 2021). FCNNs however are mappings between finite-dimensional spaces and learn mappings for single equation instances rather than learning the PDE solver. In our case FCNNs only learn mappings on fixed spatial grids. We leverage the recently formulated neural operators to extend the formulation to arbitrary grids. The key distinction is that the FCNN learns a parameter-dependent set of weights, $\Phi_{a_y}$, that has to be re-trained for every new parameter setting. The neural operator is a learned function mapping with parameter-independent weights, $\Theta$, that takes parameter settings as input and returns a function over the spatial domain, $G_\Theta(a_y)$. In comparison, the forcing term is approximated by an FCNN as $\hat{f}_{x,\Phi}(x_k; a_y) = g_{\Phi_{a_y}}(x_k)$ and by a neural operator as $\hat{f}_{x,\Theta}(x_k; a_y) = G_\Theta(a_y)(x_k)$. The mappings are given by:

$$\text{FCNN: } g_{\Phi_{a_y}} : \ D_x \to \mathbb{R}^{d_X},$$
$$\text{NO: } G_\Theta : \ H_{a_y}(D_x; \mathbb{R}^{d_{a_y}}) \to H_X(D_x; \mathbb{R}^{d_X}). \tag{17}$$

$H_{a_y}$ is a function space (Banach) of PDE parameter functions, $a_y$, that map the spatial domain, $D_y$, onto $d_{a_y}$ dimensional parameters, such as ICs, BCs, parameters, or forcing terms. $H_X$ is the function space of residuals that map the spatial domain, $D_x$, onto the space of $d_X$-dimensional residuals, $\mathbb{R}^{d_X}$.

