# OpenReview forum: "Multiscale Neural Operator: Learning Fast and Grid-independent PDE Solvers"
_ICLR.cc/2023/Conference — Submitted to ICLR 2023_

### Official Review · Reviewer_HHZo · 2022-10-15

**Confidence:** 5
**Correctness:** 2
**Technical Novelty And Significance:** 2
**Empirical Novelty And Significance:** 2
**Recommendation:** 3

**Clarity, Quality, Novelty And Reproducibility:**

The write-up is clear, the one presented experiment is described in enough detail. The idea of using neural networks to correct coarse scale numerical models is not extremely novel.

**Strength And Weaknesses:**

Learning multiscale PDE surrogates is an important direction, and this paper presents an interesting approach. However, at the current stage the paper is far away from publication. I want to address the three major issues of this paper, i.e. (i) over-ambitious claims, (ii) lack of experiments, and (iii) expensive need for training data.
Add i) In e.g. [1], Fourier Neural Operators are ablated on various operator learning problems. For the claims “grid-independent”, “flexible”, “non-local parametrizations” which are repeated mantra-like over and over again, I would have expected discussions of similar problems in this paper. Or even better, have experiments in that direction. Fourier Neural Operators in their current implementation are not really grid-independent, although claimed heavily. They can interpolate between various regular (!) grids, and I am expecting the same is true for MNO? Why is non-local parametrization so important after all? U-Nets e.g. do very well in computer vision.
Add ii) I honestly don’t think that 1d problems are really the place where multiscale PDE surrogate are going to shine. I would have expected large scale Navier Stokes problems, shallow water equations, or thereof. But ok, even if staying in 1 dimension there are many more equations, such as Kuramoto Sivashinsky or Korteweg-de Vries which are very interesting, and why not compare against the methods discussed in the related work, e.g. [2]. One experiment on one dataset is pretty scarce.
Add iii) The solver might be quasi-linear on the test set, but it still requires many many trajectories which are generated by the very same slow classical solvers the new solver should replace. That is a common chicken-egg problem in neural PDE surrogates, but especially when talking about multiscale and high resolution, I would have expected that this is addressed. That is after all a fundamental limitation.

1)	Lu, Lu, et al. "A comprehensive and fair comparison of two neural operators (with practical extensions) based on fair data." Computer Methods in Applied Mechanics and Engineering 393 (2022): 114778.
2)	Um, Kiwon, et al. "Solver-in-the-loop: Learning from differentiable physics to interact with iterative pde-solvers." Advances in Neural Information Processing Systems 33 (2020): 6111-6122.

**Summary Of The Paper:**

This paper presents a multiscale PDE surrogate, where a coarse-grained numerical model and a neural operator are added. The neural operator corrects for the errors which the numerical model makes.
The paper promises grid independent solutions, and quasi-linear runtime complexity.


**Summary Of The Review:**

The most pressing issues of the paper are (i) over-ambitious claims, (ii) lack of experiments, and (iii) expensive need for training data. These are discussed in detail in "Strength and Weaknesses".

---

### Official Review · Reviewer_ALWF · 2022-10-25

**Confidence:** 2
**Correctness:** 3
**Technical Novelty And Significance:** 3
**Empirical Novelty And Significance:** 3
**Recommendation:** 6

**Clarity, Quality, Novelty And Reproducibility:**

**Questions related to the clarity of writing**:

The bottom of Page 4 "can be written in the iterative, explicit, symbolic form... encodes the known physical equations" made me think that the equations need to be known, but it's clear from the top of page 6 that at least the high-resolution equations do not need to be known. Could you make this more clear? Is it important that the data comes from a simulation with equations of this form? Do the large-scale equations need to be known, or do you just need access to their solver? I am thinking of complicated climate models with many interacting components.

The use of lower-case x on page 6 is confusing me. Is it an index?

It seems that Appendix A.1 is unnecessary. From the main body: "We demonstrate the first steps towards quasi-geostrophic turbulence and Rayleigh-Benard convection in Appendix A.1 and aim towards integration..." In the appendix: "We are planning to demonstrate multiscale neural operator a high-dimensional system, specifically...") What does this add? Are the first steps deciding on the problem and particular PDE parameters? (Turbulence is not my expertise, so maybe this section contains some steps toward solving the problem, and I just don't understand.)

Similarly, it seems that Appendix A.2 could be shortened ("The proposed multiscale neural operator can also be leveraged for systems without explicit multi-scale formulation. We demonstrate this by formulating the MNO equations...") I could see leaving this in as a demonstration of how to formulate such a problem, but there's a lot of detail.

"Note that the system is chaotic and small deviations are rapidly amplified; even inserting the exact parametrizations in float32 instead of float64 quickly diverges." Does this mean "Trad. Param." is in float32, or is it just a comment? Does MNO use double precision? What about the "ML-based parametrization"? If the precision differs, it would be helpful to distinguish between gains from using single vs. double and gains from learning.

The axes are hard to read on Figure 4 (left) and Figure 5(a).

"The quantitive comparison of RMSE and a mean/std plot Fig. 4..." There is no mean/std plot in Fig. 4?


**Questions related to quality/correctness**:

Training & testing data are mentioned, but not validation data. Was the testing data used at all for selecting hyperparameters or other decisions, or was it held out until the very end?

Are the results from training data or test data? (That is, Figures 4, 5, and 8.) It would be helpful to compare errors on training & test data to see if there is overfitting.

Figure 8 in the appendix is very hard to read. It's not clear that orange is more accurate than blue, green, and red. It also isn't referenced. This figure (averaging error over i = 0...N but fixing k = 0, up to t=2) is much less impressive than Figures 4 & 5, and it's not clear that MNO is most accurate. Figure 4 (left) is more impressive and is for k=0, i=6, t up to 3. Is i = 6 a particularly good case? Figure 4 (right) I believe includes an average over all i, k, but I'm not sure about the length of time in terms of t (200 steps x 0.005  = 1, so maybe t = 1). Then Figure 5 (left) is X_0 (k=0, random value of i) up to t = 10. Figure 5 (right) goes up to t = 6 and is presumably averaged over all i, k. More details on these plots might illuminate the discrepancy.

"which is slightly above the optimal plateau given by the climatology": This looks like more than slightly, probably at least 30% higher? I'm also not convinced that the MNO line has plateaued.


**Strength And Weaknesses:**

To my knowledge, it is novel to use FNO for a closure problem, and it seems like a nice way to handle multi-scale physics, creating a hybrid model. I appreciate that due to the FNO approach, this method is grid-independent. I don't often see runtime complexity plots in papers like this, and I appreciate it. I also thought that the authors did a nice job of mentioning how other research could be complementary to this approach and incorporated in future work. The results are mostly impressive compared to other parameterizations, although I have questions below about the results in Figure 8. I have some questions and comments below to improve the clarity of the writing and to check on some concerns related to quality/correctness.

**Summary Of The Paper:**

This paper is about a new method named MNO (Multiscale Neural Operator), which is related to the Fourier Neural Operator. It is used to learn a closure term for a chaotic system, a parameterization for the small-scale dynamics of the multi-scale Lorenz system. The dynamics can then be propagated forward by using the solver for the large scales and the surrogate for the small scales. The method is quasilinear (much faster than the full simulation), grid-independent and non-local (via FNO), and doesn't require having a differentiable solver.

**Summary Of The Review:**

Overall, I think that this is a novel approach that might be quite effective. It heavily relies on the prior work of the Fourier Neural Operator, but incorporates some other ideas and uses it for a different task. It would have been nice to see more extensive empirical results. I have quite a few clarifying questions above, but I think that the authors will be able to address them.

---

### Official Review · Reviewer_PmjN · 2022-10-28

**Confidence:** 4
**Correctness:** 4
**Technical Novelty And Significance:** 2
**Empirical Novelty And Significance:** 2
**Recommendation:** 5

**Clarity, Quality, Novelty And Reproducibility:**

The writing is clear but can be more compact. The numerical experiment is not very complete. The novel is not very significant since there exist many similar works that consider learning closure.

**Strength And Weaknesses:**

Strength: the paper raises a good point to only learn the closure using machine learning. The hybrid model usually inherits the stability of well-studied numerical solver, but faster. Further, the paper considers scale separation, potentially making the model more efficient.

Weakness: it seems to the reviewer that there already exists a huge body of work studying hybrid models of the form ut = N(u) + K(u). For example, this paper (https://arxiv.org/pdf/2107.06658.pdf) takes a very comprehensive study of chaotic systems including Lorenz 96. Similarly, the hybrid framework that learns ML-closure has been studied on LES (https://arxiv.org/pdf/2010.10491.pdf) and climate simulation (https://agupubs.onlinelibrary.wiley.com/doi/full/10.1029/2022MS003105). The reviewer is not very sure about the novelty of the proposed framework.

Besides, the reviewer feels the numerical experiment is relatively weak. It will be more interesting to compare against standard FNO mode, or some ML-enhanced models, for example, these correcting coarse-grid simulations as discussed in the related works section.

**Summary Of The Paper:**

The paper studies a neural operator-based surrogate model, called multi-scale neural operator. It keeps the known physics knowledge, but injects machine learning to learn the closure term. The paper studies the Lorenz 96 system.

**Summary Of The Review:**

While the paper brings some interesting ideas, the experiment can be more complete. I think this work is marginally below the threshold for acceptance.

---

### Official Review · Reviewer_rbkc · 2022-11-03

**Confidence:** 4
**Correctness:** 3
**Technical Novelty And Significance:** 2
**Empirical Novelty And Significance:** 2
**Recommendation:** 3

**Clarity, Quality, Novelty And Reproducibility:**

Clarity and Quality

The paper is overall well written but not necessarily pedagogical or scientifically step-by-step. One important confusion that it perpetuates is that the Fourier Neural Operator is something other than a convolutional network.Further, mathematical notation about operators in Banach spaces is introduced but never used. Finally, the narrative around climate change sandwiching the paper is unfortunately not really helpful, since no real climate science is done,  and it takes up valuable space that could have been filled with other information, especially more diagnostic experiments, or more real-data analyses.

Novelty
The issue with the novelty claim in this paper is that it hinges on the idea that the Fourier Neural Operator is in some way distinguished and not comparable to a standard convolutional network because it is said to be "grid-independent" and "non-local". It is worth repeating that underlying both FNO and standard convolutional networks are interpolation schemes, and none is more grid-independent than the other. Removing this conceptual barrier makes the novelty claim tenuous since closure/parameterization methods with CNNs exist (and are cited) and not compared to.

Reproducibility
The authors have announced that code will be released. In this sense I do not doubt the possibility of reproducing the experiments done.

**Strength And Weaknesses:**

Strengths:
- The proposed method clearly shows a significant improvement in speed and accuracy in the bench marks- There is an extensive related work section- Advancing climate science is a very important goal

Weaknesses:
Work on improving climate simulations is very important and I am all for pushing this forward at full power. Alas, there are a number of weaknesses to this contribution that give me pause concerning its utility and fitness for publication at ICLR.

- It is very important to note that the Fourier Neural Operator is not actually grid-independent, or alternatively to note that it is precisely as grid independent as all convolutional methods based on stencils in signal space. The only difference is that they rely on slightly different interpolation kernels. This is a very important point, and of course the original FNO paper is not up for discussion here, but this misconception should be avoided in papers that reference the FNO paper in order to avoid further confusion.

- The reason the above note is very important, is that it removes a number of the perceived differences between the FNO and other convolutional approaches, for example that in LaPeyre 2019, or Bar-Sinai 2020 (not cited). The main difference that remains to other convolutional approaches is the possibility of having filters that are larger in size than the usual 3x3 or 5x5 stencils. A number of other convolutional methods must hence be compared to assess which type of model is best for closure modeling. Further, the UNet approach in Lapeyre 2019, is actually substantially non-local and actually as global as required. Similarly, for speed, it would be great to have reference benchmarks of classical methods that use non-learned closure models.

- It is entirely unclear to what extent the inductive bias of the FNO, which is a convnet that can allow large-scale convolutional filters, is actually required in the application tested here. It would be great to see tests on this by progressively restricting the possible convolutional filter size and see its impact on different problem settings.

- It is not explained how the Fourier Neural Operator is adapted to the current problem (was it modified? If so, how?).

- In general, it would be great to have a detailed section with toy examples. Beyond the Lorenz 96 system, it would be great to see a) transport equation. what happens to a transported dirac? b) wave equation. It is possible to build a wave equation to which the solution is a superposition of two waves, one high-frequency, one low-frequency (use a forcing term or involve the zeroth derivative in the equation). Make the coarse-graining a strict low-pass allowing only one wave through. How does the closure model behave in this non-interacting setting? c) A simple phase-amplitude coupling model should be explored (phase of a low-frequency oscillation dictates amplitude of a high-frequency oscillation). d) Simple diagnostics on Burgers' (shocks are multi-frequency) and KS (what happens at different smoothing scales?). For all of these examples, it would be great to see the contribution of the coarse-grained model and the correction from the learned part.

- Finally, it would be great to see this method in action on some real, relevant data. This can be either from climate science or astrophysics or other hydrodynamic applications with turbulence and multi-scale effects. It may be interesting to compare bottom-up 2D turbulence to top-down 3D turbulence. A comparison to Stachenfeld 2022 on the data set that they use would be adequate.

- (minor) The proposed method is not multi-scale, but at best two-scale. The Russian doll figure seems to insinuate that there are multiple scales involved or that the method can be applied recursively, but that does not seem to be the case here. This makes the naming quite misleading. It would be much better to name it something related to closure, or make it actually multi-scale, since the current discrepancy will cause confusion.


**Summary Of The Paper:**

This paper introduces the Multi-scale Neural Operator for ML-based closure/parameterization of PDEs with multi-scale properties. In the framework of spatial coarse-graining of PDEs, the multiscale neural operator relies on a fast/easy/simple solve of a coarse-grained problem and estimates/computes the commutator between the coarse-graining and the spatial differential operator by learning from data. Obtaining this commutator accurately allows one to run a differential equation solve entirely in coarse-grained space, while incorporating the relevant information from the smaller subgrid scales.The context of weather simulations is chosen as an example application, the improvement of which can have real-world impact in combating climate change. It is shown on the Lorenz96 model that the proposed solver is significantly faster than direct solves at high resolution, and significantly more accurate than selected competing methods.

**Summary Of The Review:**

This paper introduces the Multiscale Neural Operator for closure/parameterization problems. It proposes to learn the closure commutator.

While the stated goals of improving climate science and combating climate change are noble, and the numerical results look convincing, many required comparisons are missing, which may better situate the performance of the proposed method. It is further suggested to analyze and visualize the behavior of the method on a larger set of toy problems.
Finally, it is pointed out that the term Multiscale Neural Operator is not very descriptive of the procedure.

I do hope the authors are willing and able to address some of these comments and am open to changing my assessment if that is the case.

---

### Decision · Program_Chairs · 2023-01-20

**Decision:**

Reject

**Justification For Why Not Higher Score:**

Writing not adapted  for an ML audience, technical imprecisions, limited experiments, lack of baselines.

**Justification For Why Not Lower Score:**

N/A

**Metareview: Summary, Strengths And Weaknesses:**

The paper presents a model integrating a neural operator and a PDE solver for modeling multiscale systems: the former is used to learn the closure terms (small-scale dynamics) that influence large scale dynamics modeled by the coarse-grained solver. Training is performed by using high fidelity data. Experiments are performed on the Lorenz96 chaotic system.

The paper presents an original solution for solving closure problems, an important issue in many geophysics fields. It heavily builds on Fourier neural operators, but presents some interesting novelties. In its present form it is however not amenable for publication. The organization and writing are not adequate for a ML conference and will probably benefit only to a small audience. The reviewers highlight over-claims and imprecisions. The experiments have been performed on only one dynamical system and there is a lack of comparison with related approaches. Overall the paper will benefit from a rewrite and a strengthening of the experiments.
The authors did not post any rebuttal.